# Ectopic Expression of *Os-miR408* Improves Thermo-Tolerance of Perennial Ryegrass

**Geli Taier** [1]**, Nan Hang** [1]**, Tianran Shi** [1]**, Yanrong Liu** [2]**, Wenxin Ye** [3]**, Wanjun Zhang** [1,4,*] **and Kehua Wang** [1,*]

1    College of Grassland Science and Technology, China Agricultural University, Beijing 100193, China; taiergeli@cau.edu.cn (G.T.); nhang1997@163.com (N.H.); shitianran123@163.com (T.S.)
2    College of Biological Sciences, China Agricultural University, Beijing 100193, China; liuyanrong142726@163.com
3    College of Grassland, Resources and Environment, Inner Mongolia Agricultural University, Huhehaote 010011, China; ywxing07@163.com
4    National Energy R&D Center for Biomass, China Agricultural University, Beijing 100193, China
*    Correspondence: wjzhang@cau.edu.cn (W.Z.); kehwang@cau.edu.cn (K.W.)

**Abstract:** With global warming, high temperature stress has become a main threat to the growth of cool-season turfgrasses, including perennial ryegrass. As one of the conserved plant microRNA families, miR408s are known to play roles in various abiotic stresses, including cold, drought, salinity, and oxidative stress, but no report, thus far, was found for heat. Here, perennial ryegrass plants overexpressing rice *Os-miR408* were used to investigate the role of miR408 in plant heat tolerance. Both wild type (WT) and miR408 transgenic perennial ryegrass plants (TG) were subjected to short-term heat stress at 38 °C for 72 h (experiment 1) or at 42 °C for 48 h (experiment 2), and then let recover for 7 days at optimum temperature. Morphological changes and physiological parameters, including antioxidative responses of TG and WT plants, were compared. The results showed that miR408 downregulated the expression of two putative target genes, *PLASTOCYANIN* and *LAC3*. Additionally, overexpression of *Os-miR408* improved thermo-tolerance of perennial ryegrass, demonstrated by lower leaf lipid peroxidation and electrolyte leakage, and higher relative water content after both 38 and 42 °C heat stresses. In addition, the enhanced thermotolerance of TG plants could be associated with its morphological changes (e.g., narrower leaves, smaller tiller angles) and elevated antioxidative capacity. This study is the first that experimentally reported a positive role of miR408 in plant tolerance to heat stress, which provided useful information for further understanding the mechanism by which miR408 improved plant high-temperature tolerance, and offered a potential genetic resource for breeding heat-resistant cool-season turfgrass in the future.

**Keywords:** miR408; heat stress; perennial ryegrass; antioxidant enzymes; morphological changes





## 1. Introduction

Environmental stress, including heat, is one of the major factors restricting agricultural production worldwide. Statistical results show that global environmental stress causes crop yields to be reduced by over 50% each year, and every increase in temperature by 1 °C will directly cause crop yield losses of 2.5–16% [1]. More seriously, due to the global warming caused by greenhouse gas emissions, the global average temperature may rise by 2–4.5 °C in this century, and the temperature in some areas may even rise by 6 °C [2]. Therefore, it is very important to find ways to enhance heat stress resistance of crop plants, including cool-season turfgrasses.

Perennial ryegrass (*Lolium perenne* L.) is a widely grown cool-season turf and forage grass species in the world. It has many desirable agronomic traits, including dark green color, rapid growth and establishment, good turf quality, and high yield as a forage grass under favorable growth and environmental conditions. Nevertheless, as a perennial cool-season grass, it performs best at a temperatures ranging from 15 to 25 °C, and the growth

usually begins to decline as the temperature goes beyond 27 °C. It cannot withstand high-temperature weather, and heat stress is a main factor limiting its growth and development in transition and warmer regions [3].

Plant microRNAs (miRNAs) are a group of small non-coding RNAs at the length of about 19 to 25 nucleotides. They function in post-transcriptional regulation of gene expression and RNA silencing via base pairing to complementary sequences within target mRNA molecules [4]. As a vital and evolutionarily conserved component of gene regulation, miRNAs play important and various roles in plant development, such as floral organ identity, leaf morphogenesis, and seed development; and many of them are involved in responses to different abiotic and biotic stresses, including miR408 [5]. MiR408 is one of the conserved miRNA families in land plants, and researchers have showed its participation in many different developmental processes, including photosynthesis, vegetative growth, flowering time, and yield [6,7]. Moreover, miR408 has also been reported to participate in diverse environmental stresses, such as iron deficiency [8], light and copper [9], oxidation [10], salt [10], drought/osmotic stress [11], and low temperature [10,11]. MiR408 conservatively targets transcripts for copper-binding proteins, such as phytocyanins and laccases (*LAC3*, *LAC12*, *LAC13*), as confirmed by cleavage site analysis in *Arabidopsis* [12]. For example, miR408 functions in regulating grain yields and photosynthesis [6] and in adjusting various stress responses in diverse plant species [9,10,13] by targeting phytocyanins (e.g., *UCL8*, *PLANTACYANIN1*, and *PLASTOCYANIN*) and/or laccases. In addition, miR408 can regulate some non-conserved target genes, such as timing of CAB expression 1 (*TOC1*), also known as *PRR1*, pseudo-response regulator 1) for heading time in wheat [7].

Recently, the expression of *miR408* was reported to be induced by heat in herbaceous peony (*Paeonia lactiflora* Pall.) with a much higher level in 'Zifengyu', a thermo-tolerant cultivar, suggesting the possible involvement of miR408 in heat stress responses and heat tolerance [14]. Similarly, *miR408* expression increased under high temperature in celery (*Apium graveolens* L.) [15]. Contrarily, a study in switchgrass found that heat downregulated the expression of *miR408* [16]. In a genome wide profiling study of miRNAs in rice, *miR408* expression levels were decreased in the heat tolerant cultivar N22, but increased in the susceptible cultivar Vandana [17]. Other than the contradictory observations of miR408 in response to high temperature among different plant species, experimental evidence is lacking on the function of miR408 in plant heat stress. The objective of this study was to examine the biological function of miR408 in heat stress using previously established transgenic perennial ryegrass over-expressing the rice *Os-miR408* gene [18]. Knowledge generated in this study will contribute valuable information towards understanding the role of miR408 in plant responses to heat stress, perhaps leading to breeding more thermo-tolerant perennial ryegrass for transitional and southern regions.

## 2. Materials and Methods

### 2.1. Plant Maintenance and Heat Stress Treatment

Experiments were performed in perennial ryegrass "Citation Fore" (Pure Seed, Canby, OR, USA). Transgenic plants overexpressing *Os-miR408* were generated as described previously [18]. Wild type (WT) control plants and three lines of transgenic perennial ryegrass plants (TG1-1, TG5-2, TG7-1) were clonally propagated from tillers in plastic pots (8.5 cm depth and 10 cm diameter) filled with a soil mixture of vermiculite: sand: peat (1:1:1) in the greenhouse (15 ± 2 °C/25 ± 3 °C, night/day) from May to October 2019 at China Agricultural University (Beijing, China). The plants of the same growth status were selected and transferred to a plant growth chamber (PQX-450, Haixiang Instrument and Equipment Company, Shanghai, China) provided fluorescent white light, photosynthetically active radiation(PAR) of 450 $\mu$mol s$^{-1}$ m$^{-2}$, and a dark/light cycle of 16/22 °C, 10/14 h, and 70% relative humidity. Watering took place every two days. In order to minimize the effects of the microenvironment, the pots in the chamber were rotated every 24 h. The plants were clipped to 10 cm and fertilized once a week with Miracle-Gro (N-P-K 24-12-14, Scotts, Wuhan, China) at a rate of 5 kg N ha$^{-1}$ to achieve uniform plant growth.

After acclimatizing to chamber conditions for 2 weeks, plants were randomly subjected to either a 72h (38/38 °C, 14/10 h, day/night, PAR at 450 μmol s$^{-1}$ m$^{-2}$) (experiment 1) or 48h heat stress treatment (42/42 °C, 14/10 h, day/night, PAR at 450 μmol s$^{-1}$ m$^{-2}$) (experiment 2). Four replicates were prepared for WT and each three transgenic lines. During the heat stress, a beaker with water was placed in the plant growth chamber to ensure that the water's temperature was the same as the air temperature in the chamber. The plants were watered every 12 h to keep the soil moisture sufficiently and observed regularly to ensure that there was no wilting symptom. After 72 or 48 h of high-temperature treatment, normal growth conditions were restored and the plants were allowed to recover for seven days. To observe regrowth after stress, all of the grass plants were cut at about 4 cm before returning to normal growth conditions for recovery.

*2.2. Sampling and Measurements*

2.2.1. Phenotypic Analysis of TG Plants

Phenotypic assessment was conducted on 5-month-old WT control and TG plants maintained under normal growth conditions. The width of the plant leaves was determined by measuring the width of the widest part of the first fully developed leaf (from the top to the bottom of the plant) of each tiller with a vernier caliper. Fifteen measurements were taken for each plant line. Those samples were also used to observe the cross-section of the leaves. Tiller angles were measured between the tiller and the vertical on 3 randomly selected outside tillers of each pot [19]. After seven days of recovery, new leaves were counted for both WT and TG plants. For thermo-imaging, the detached leaves of WT and TG plants were heat-treated in a 42 °C incubator for 10 min, and an infrared camera (Seek Thermal XR Imager, Seek Thermal, Goleta, CA, USA) was used to take pictures immediately after removing the leaves from the incubator.

2.2.2. RNA Extraction and Expression Analysis

The total leaf RNA was isolated from WT and TG plants (TG7-1, TG5-2, and TG1-1) using the TRIzol reagent (Invitrogen, Carlsbad, CA, USA) before stress treatment, and then treated with RNase-free DNase I. First-strand cDNA was synthesized with Invitrogen SuperScript III Reverse Transcriptase Kit following the product manual. Semi-quantitative reverse transcription PCR (RT-PCR) was conducted for *LAC3* and *PLASTOCYANIN* with specific primers (Table S1). A 25-μL reaction volume was used for the RT-PCR, containing 1 μL cDNA template, 1 μL of each primer, 22-μL reaction Mix (SYBR Green I). The thermal cycler amplification condition was 98 °C for 2 min; 98 °C for 10 s, 60 °C for 10 s, and 72 °C for 10 s with 35 cycles; 72 °C for 5 min, then cool down to 4 °C. Stem-loop RT of a 25-μL reaction volume was performed according to a previous protocol [20] with primers (Table S1). *ACTIN* gene (Table S1) was adopted as the internal reference. The thermal cycler amplification condition was 98 °C for 2 min; 98 °C for 10 s, 58 °C for 10 s, and 72 °C for 10 s with 35 cycles; 72 °C for 5 min, then cool down to 4 °C.

2.2.3. Measurement of Physiological Parameters

Chlorophyll quantification was conducted as described previously [21]. Fully expanded leaves were harvested at 0 and 48 h after 42 °C heat stress treatment, cut into pieces, and then soaked in 95% ethanol in the dark until turning white. The absorbance was read at 665 and 649 nm. Chlorophyll a/b was calculated as milligrams per gram fresh weight with the following equations: chlorophyII a = $13.95 \times (A_{665}) - 6.88 \times (A_{649})$; chlorophyII b = $24.96 \times (A_{649}) - 7.32 \times (A_{665})$. Leaf maximum photochemical efficiency of photosystem II (PSII) (Fv/Fm) was measured after a 20-min dark adaptation with OS-30P chlorophyll fluorometer (Opti-Sciences, Tyngsboro, Hudson, NH, USA) at 0 and 32 h after 42 °C heat stress treatment.

Leaf relative water content (RWC) was measured in accordance with the method by Barrs and Weatherley [22], with modification. Fully expanded leaves were harvested at 0 and 72 h after 38 °C treatment or 0 and 48 h after 42 °C treatment. Ten fully expanded

leaves (weighed about 0.2 g) from each pot were weighed (W1) and cut into 1–2 cm pieces. The weight of the leaves after 24 h of water absorption was measured (W2). The leaves were then dried to constant weight (W3) in an oven at 65 °C. Relative water content (%) = (W1−W3)/(W2−W3) × 100%.

Electrolyte leakage (EL) was measured as described by Blum and Ebercon [23], with some modifications. Fully expanded leaves of 0.2 g were harvested at 0 and 72 h after 38 °C and 48 h after 42 °C treatment, and then soaked into 20 mL deionized distilled-water. The initial conductivity (S1) was measured with a conductivity meter (Hengxin TDS86555, Taiwan) after shaking the leaves on a shaker for 24 h. The final conductivity (S2) was determined after killing the leaves in an autoclave for 20 min and then cooling to room temperature. EL = S1/S2 × 100%.

Lipid peroxidation of the leaf tissue was measured in terms of malondialdehyde (MDA) content by thiobarbituric acid colorimetry, with some modifications [24]. The leaves were harvested at 0 and 48 h after 42 °C treatment, and 0.2 g were ground in 10% (*v/v*) trichloroacetic acid. A 2-mL aliquot of the supernatant was mixed with 2 mL 0.6 % (*v/v*) thiobarbituric acid. The mixture was heated at 100 °C for 15 min, quickly cooled in ice water, and then read at 532 and 600 nm with a UV–VIS spectrophotometer (Hitachi UH5300, Tokyo, Japan). The value for the non-specific absorption at 600 nm was subtracted from the absorption value at 532 nm. The MDA concentration was calculated by means of an extinction coefficient of $155\text{-mM}^{-1} \text{ cm}^{-1}$.

Enzyme extracts were prepared using the method by Chaitanya et al. [25], with modifications. Leaf samples were taken at 0 and 48 h after 42 °C treatment, and 0.2 g leaves was frozen quickly in liquid N and then ground in 4 mL solution containing 50 mM pH 7.8 phosphate buffer, 0.1 mM ethylenediaminetetraacetic acid (EDTA), and 1% polyvinylpolypyrrolidone. The homogenate was centrifuged at 4 °C, $10,000 \times g$ for 20 min, and the supernatant was used for enzyme activity assays.

The activities of superoxide dismutase (SOD), ascorbate peroxidase (APX), guaiacol peroxidase (POD), and catalase (CAT) were determined following a previous protocol [25]. A 3 mL assay mixture for measuring SOD enzyme activity contained 50 mM phosphate buffer (pH 7.8), 75 μM nitro blue tetrazolium chloride (NBT), 2 μM riboflavin, 13 mM methionine, 0.1 mM EDTA, and 0.1 mL enzyme extract. The mixture was then illuminated at 80 to 100 $\mu$mol m$^{-2}$ s$^{-1}$ for 10 min. The reaction mixture without enzyme extract developed the maximum color after illumination and was used as maximum reduction of NBT, and the mixture without illumination served as the control. The absorbance at 560 nm was measured with a UV–VIS spectrophotometer. For measuring APX enzyme activity, a 3 mL reaction mixture contained 50mM pH 7.0 PBS, 0.1 mM $H_2O_2$, 0.5mM ASA, and 100 μL enzyme extract. The absorbance was recorded at 290 nm for 1 min. A 3 mL assay solution for measuring POD enzyme activity contained 20 mM phosphate buffer (pH 6.0), 0.4 mM 2-methoxyphenol, and 0.3 mM $H_2O_2$, and 200 μL enzyme extract. The absorbance was measured at 470 nm for 1 min. The activity of CAT was determined by mixing 3 mL reaction solution containing 50 mM PBS (pH 7.0), 10 mM $H_2O_2$, and 0.1 mL enzyme extract. The absorbance was measured at 240 nm for 1 min.

Leaf hydrogen peroxide ($H_2O_2$) contents were detected according to Velikova et al. [26] with some modifications. The samples were harvested at 0 and 48 h after 42 °C treatment. Likewise, 100 mg of leaf tissues was frozen in liquid nitrogen and ground into fine powder, and then mixed well with 1.5 mL 0.1% (*w/v*) trichloroacetic acid. The mixture was centrifuged for 15 min at $12,000 \times g$ and 4 °C. Total 0.5 mL supernatant was mixed with 0.5 mL 10 mM pH 7.0 potassium phosphate buffer and 1 mL 1 M KI. The absorbency of the mixture was recorded at 390 nm. The $H_2O_2$ content was calculated based on a standard curve.

### 2.3. Experimental Design and Statistical Analysis

The experiment was a randomized complete block design with either four replicates (experiment 1, 4 pots for each line of plants) or five replicates (experiment 2, 5 pots for each line of grass plants). In this study, the high temperature was not regarded as a factor of

treatment, but more of an abiotic stress to which all grass plants were exposed uniformly to test the performance difference, if any, between the WT control and TG plants under heat. The two heat stress experiments were analyzed separately. We analyzed all the measurements using the samples collected at the sampling time described previously (before and 72 h after heat stress, or before and 48 h after heat stress), unless stated otherwise. All of the data obtained were analyzed using SPSS software (Version 25.0, IBM, Armonk, USA). Treatment means were separated using the Fisher's least significant difference (LSD) test at a significance level of 0.05.

## 3. Results and Discussion

### 3.1. Over-Expressing miR408 Plants Have Narrower Leaves and Smaller Tiller Angles

To determine the phenotypic characteristics of over expressing *miR408* plants, we observed the WT and transgenic plants are grown for 4 weeks in growth chambers under 22°C/16°C conditions (Figure 1). Compared with the WT plants, the TG plants had finer leaves (Figure 1A,C,D) and grew more erectly with smaller tiller angles (Figure 1E–I).

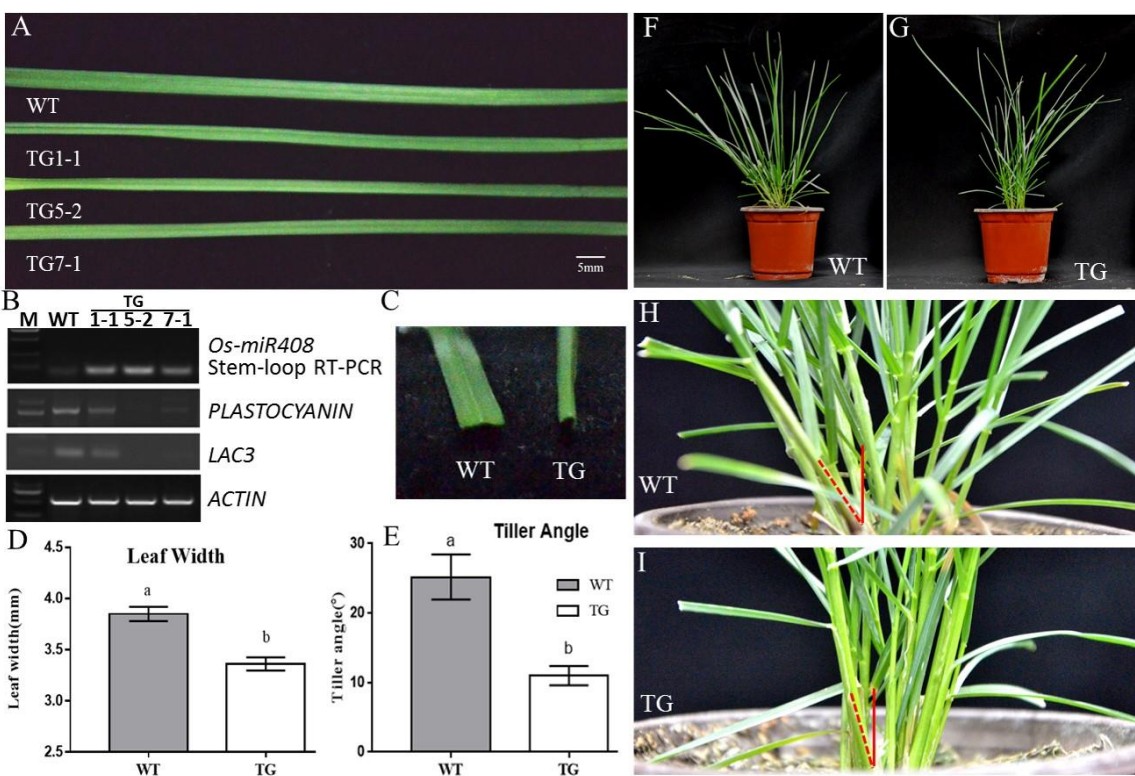

**Figure 1.** Phenotypic analysis and putative target gene expression analysis of perennial ryegrass overexpressing *Os-miR408*. (**A**) Comparison of leaf width of wild type control (WT) and transgenic plants (TG). (**B**) Expression of matured *miR408* and two putative target genes, *PLASTOCYANIN* and *LAC3* in WT and three TG lines by semi-quantitative RT-PCR. (**C**) Cross-sectional images of WT and TG plant leaves. (**D**) Leaf blade width; (**E**) tiller angle between representative WT and TG plants. (**F**,**G**) Comparison of WT and TG plant architecture, and (**H**,**I**) comparison of WT and TG plant tiller angle (the angle between the tiller and the vertical, as shown by the red dash line and solid line). Error bars represent standard error (*n* = 15). The different lower-case letters indicate a significant difference between WT and TG plants at *p* < 0.05.

Extensive studies were conducted on the tiller angle, and several genes that are involved in the control of tiller angle were identified and cloned, including *LAZ1*, *PROG1*, *TAC1*, *TAC3*, *LPA1*, and *TIG1* [19,27,28]. More recently, miR167 was reported to control the tiller angle by targeting auxin response factors and affecting auxin distribution [19]. It was suggested that miR408 is involved in anthocyanin biosynthesis through auxin-mediated signaling [29]. Auxin responsive Aux/IAA gene (Os01g53880) was predicted as an atypical

target of miR408 in cold response [11]. Nevertheless, whether and how miR408 might affect tiller angle via auxin-mediated signaling pathway is unknown, and further studies are warranted. Tiller angle is a very important morphological character that influences plant architecture. Plants with smaller tiller angles or more erect growth habit often have a compact plant architecture, which might increase plant density, enhance photosynthetic efficiency, and eventually enhance plant performance [19]. In addition, more upright plant architecture (increased leaf hyponasty, smaller tiller angle) is a plant thermomorphogenic phenomenon for plants under higher than optimum temperature. Finer and smaller leaves are also thought to be a morphological adaptation for high temperature [30]. Thus, we suggested that the morphological changes observed in *miR408*-over-expressing plants might enable the TG plants to better adapt to high temperatures.

### 3.2. Genes Encoding PLASTOCYANIN and LAC3 Were Downregulated in miR408 Over-Expressing Plants

We predicted the miR408 target genes in perennial ryegrass by psRNATarget (http://plantgrn.noble.org/psRNATarget) using the transcriptome data previously reported [31] and further measured the expression of two putative targets, LAC3, and plastocyanin(Tables S2 and S3). As shown in Figure 1B, the expression of these two genes was repressed in all the three transgenic perennial ryegrass lines, particularly in TG5-2 and TG7-1.

The miR408-mediated downregulation of certain blue copper proteins, which regulate a range of plant development processes (e.g., vegetative growth, photosynthesis, heading time, and grain size) and environmental stress responses, has been reported as a conserved regulatory pathway across different plant species [6,10]. In Arabidopsis, miR408 targets PLANTACYANIN1 and three laccase genes *LAC3*, *LAC12*, *LAC13* [12]. In rice, *OsUCL8* and *OsUCL30* were predicted targets of miR408, and were validated by 5′ RACE [32]. Four plastocyanins identified by psRNATarget prediction were also validated as targets by investigating their expression in *Os-miR408* overexpression lines [11]. Our results are consistent with previous findings of the downregulating of *PLASTOCYANIN* and/or *LAC3* in miR408 over-expressing lines [6,10]. These indicated that the two genes could be involved in the regulation of morphological changes and heat stress responses observed in the transgenic perennial ryegrass plants.

### 3.3. Os-miR408 Over-Expressing Perennial Ryegrass Were More Tolerant to Heat Stress
3.3.1. Transgenic Plants Show Lower Morphological Damage and Maintain Higher Physiological Activities after High-Temperature Stress

In order to investigate the possible role of miR408 in plant heat stress tolerance, we imposed two types of heat stresses (38 °C for 72 h and 42 °C for 48 h) individually on both the WT control and *Os-miR408,* over-expressing perennial ryegrass plants along with recovery, for 7 days, and monitored the morphological damage and some of the physiological responses (Figures 2 and 3). After 72 h of high-temperature treatment at 38 °C, we observed significant differences between WT and TG plants, with obvious wilting and leaf burning for WT, but not for TG plants (Figure 2B,C). Under more severe high-temperature treatment at 42 °C, similar results were found 32 h after stress initiation. The leaves of the WT plants wilted, the leaf edges were scorched, and the leaves were somehow grayish (Figure 3E). However, the TG plants were less damaged, and the whole plant still showed a healthy dark green color (Figure 3E). After 48 h of high temperature (42 °C) treatment, the leaves of WT plants were almost completely dehydrated and burned, while TG plants showed about 50–60% wilting of the entire leaves, and the damaged leaves mostly distributed outside the plants. The majority of the leaves in the middle still maintained relatively normal growth, and TG1-1 and TG7-1 performed best among the transgenic plants, while TG5-2 performed relatively poorer (Figure 3B,F). Seven days after recovery, the TG plants started growing new leaves, while the WT plants nearly died (Figure 2D,H and Figure 3C,G). The infrared photograph showed that the leaf surface temperature of TG plants after heat treatment was 2–4 °C lower than that of WT control, and the hot area was significantly smaller than that of the WT, indicating a stronger heat dissipation capacity

(Figure 3H). Generally, the morphological damage observed in vascular plants in response to heat stress include branch and leaf burn, foliar abscission and senescence, inhibition of vegetative growth, discoloration and so on [33]. The degree of damage mainly depends on stress intensity, duration, and the plant resistance to heat stress [34]. The above results showed that overexpression of *Os-miR408* improved heat tolerance of perennial ryegrass, which might be partially attributed to the adaptive morphological changes in TG plants.

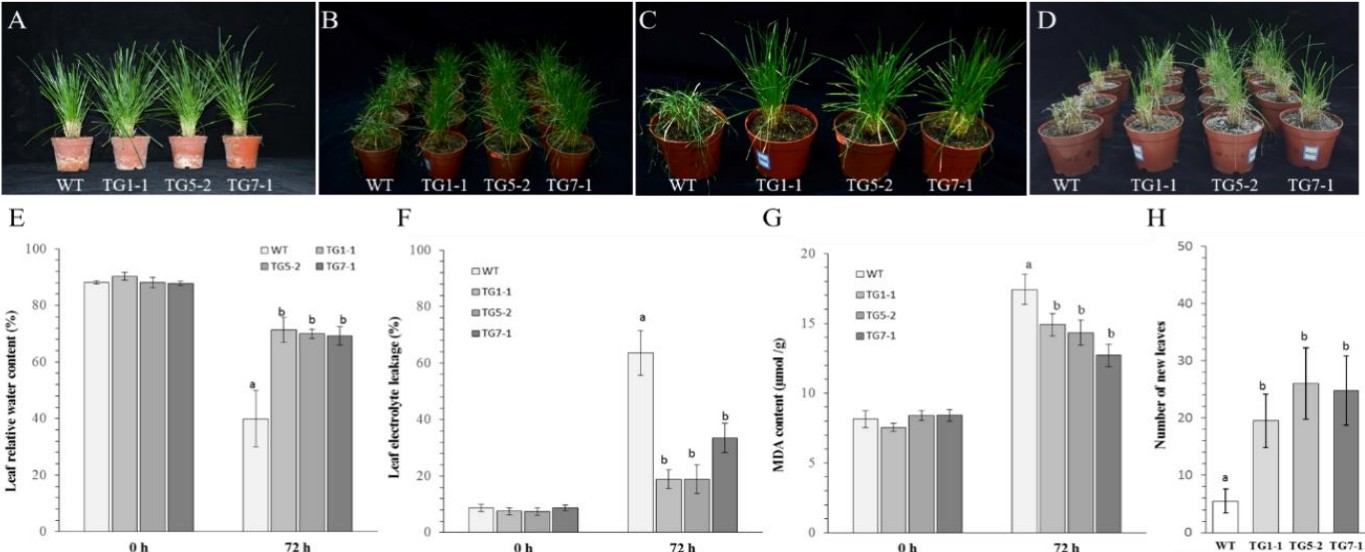

**Figure 2.** Performance of three *Os-miR408* transgenic perennial ryegrass lines (TG1-1, TG5-2, TG7-1) and wild type (WT) control under heat stress of 38 °C for 72 h in experiment 1. (**A**) WT and TG plants were under normal condition; (**B,C**) subjected to heat stress at 38 °C for 72 h; and (**D**) 7 days after recovery under normal condition. (**E**) Leaf relative water content (RWC); (**F**) leaf electrolyte leakage (EL); (**G**) malondialdehyde (MDA) content; and (**H**) the number of new leaves of WT and TG plants regrown from the cutting seven-day after the recovery. Error bars represent standard error (*n* = 4). Different lower-case letters indicate a significant difference between WT and TG plants at *p* <0.05.

High temperature causes the chlorophyll content to decrease by promoting the degradation of chlorophyll in plants, but heat-resistant varieties can maintain higher chlorophyll content than heat-sensitive ones under stress conditions. Thus, chlorophyll could be a physiological indicator for stress tolerance evaluation under high temperature [35]. Under normal growth conditions, there was no significant difference in chlorophyll content between WT and TG plants, while after high temperature treatment, the chlorophyll content of WT significantly reduced, and was lower than in all three TG lines, except TG1-1 (Figure 4A). Similarly, significant differences of Fv/Fm between WT and TG plants were found 32 h after heat stress, and it decreased by 84% in WT, but to a lesser extent in TG plants (Figure 4B). Heat stress destroys the structure of thylakoid in chloroplast, inhibits the electron transfer of PSII and greatly reduces or even stops its activity [36]. Chlorophyll fluorescence parameters can indicate the physiological conditions of photosynthesis in plants, and changes in chlorophyll fluorescence parameters under stresses can reflect plant tolerance to different abiotic stresses, including high temperature [33]. Ma et al. [10] also found the measured Fv/Fm values were very similar between 35S:miR408 transgenic Arabidopsis plants and the wild type when grown under optimal temperatures, but they were lower in the wild type plants following 12-h cold incubation compared with 35S:miR408 plants. These results showed that under high temperature stress, the chlorophyll and light energy conversion efficiency of PSII are less affected in TG perennial ryegrass overexpressing *Os-miR408*, indicating its superior capacity to resist heat stress and to maintain photosynthetic efficiencies under stress.

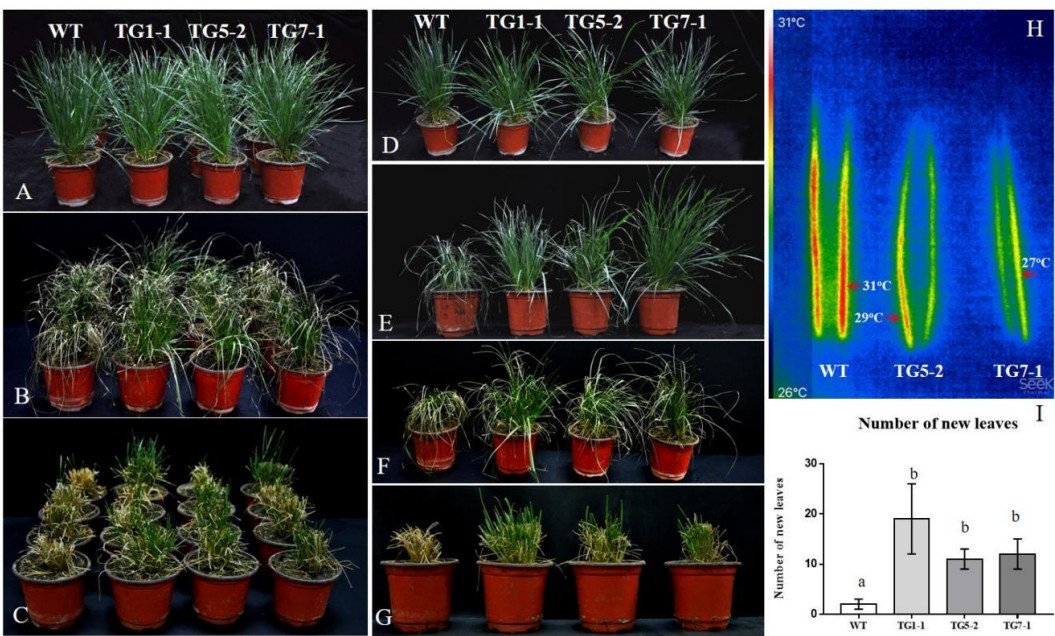

**Figure 3.** Performance of three *Os-miR408* transgenic perennial ryegrass lines (TG1-1, TG5-2, TG7-1) and wild type (WT) control under heat stress of 42 °C for 48 h in experiment 2. (**A,D**) WT and TG plants were under normal conditions; (**B,F**) subjected to heat stress at 42 °C for 48 h; and (**C,G**) 7 days after recovery. (**E**) subjected to heat stress at 42 °C for 32 h; (**H**) Infrared imaging of surface temperature of detached leaves of WT and TG plants subjected to heat shock at 42 °C for 10 min. (**I**) The number of new leaves of WT and TG plants regrown from the cutting seven-day after the recovery. Error bars represent standard error (*n* = 5). Different lower-case letters indicate a significant difference between WT and TG plants at *p* < 0.05.

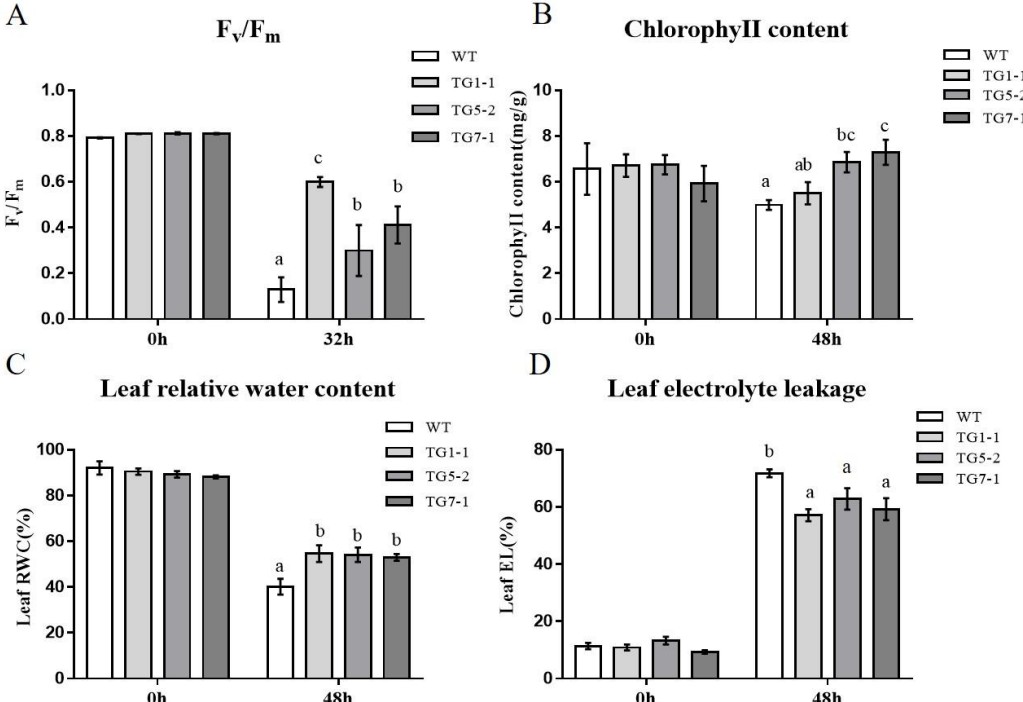

**Figure 4.** Photochemical efficiency of photosystem II (Fv/Fm) (**A**); total leaf chlorophyll content (**B**); leaf relative water content (RWC) (**C**); and leaf electrolyte leakage (EL) (**D**) of wild type control (WT) and *Os-miR408* transgenic lines (TG) before and after heat stress treatment at 42 °C for 48 h (experiment 2), except Fv/Fm, which was measured 32 h after heat stress; 0 h, before heat stress; 32 h, 32 h after heat stress; 48 h, 48 h after heat stress. Error bars represent standard error (*n* = 5). Different lowercase letters indicate a significant difference between the WT and TG plants at *p* < 0.05.

High temperature can also result in tissue dehydration, and plants with better heat resistance usually maintain higher relative water content and relatively normal cell metabolism under heat stress [33]. Under normal growth conditions, there was no difference in leaf RWC between WT and TG plants, ranging from 87.7% to 90.2%, and 88.2% to 92.1% in experiment 1 and 2, respectively. After high temperature stress, the leaves of both WT and TG plants lost water (Figures 2E and 4C). For instance, the RWC of WT decreased to 47.1% 48 h after heat stress in experiment 2, which was significantly lower than that of TG plants (Figure 4C).

In addition, heat also damages cell membrane integrity and stability, causing increase of electrolyte leakage. The ability to maintain cell membrane stability and integrity is a major indicator of heat tolerance. Compared with less heat-sensitive plants, heat-resistant plants usually have lower electrolyte leakage and stronger plasma membrane stability under stress [37]. The EL of the leaves was maintained at about 10% under normal growth conditions in both experiments, but increased significantly after being subjected to high temperature stresses, especially the WT, reaching 63.7% and 71.8%, which was higher than those of TG lines in experiments 1 and 2, respectively (Figures 2F and 4D). Similar results were reported in *miR408* overexpression transgenic plants under cold stress, with a lower EL in comparison with WT [10,11]. Compared to the WT, TG perennial ryegrass overexpressing *Os-miR408* had higher water-retention capacity and stronger cell membrane stability, suggesting that they are less susceptible to high temperature damage.

### 3.3.2. MiR408 Increased Plant Antioxidant Capacity

Heat inhibition of photosynthesis results in an imbalance of the electron-transfer chain and promotes production of reactive oxygen species (ROS, $O_2^-$, $^1O_2$, $^{\bullet}OH$, and $H_2O_2$) [38,39]. Excess ROS are detrimental, and they could cause the autocatalytic peroxidation of membrane lipids, protein oxidation, nucleic acid damage, and enzyme inhibition [33,36]. As the product of unsaturated fatty acids peroxidation within lipids, MDA is often determined and widely used to monitor such oxidative damage caused by ROS [33]. In the present study, MDA contents increased significantly after heat stress in both experiments, regardless of plant lines (Figures 2G and 5B). For example, in experiment 2, it rose to 25.2 μmol/g FW in WT plants, which was significantly higher than that of all three TG lines, except TG5-2 (Figure 5B). This result was concomitant with elevated $H_2O_2$ production after heat stress, and WT plants had much higher $H_2O_2$ production than TG plants (Figure 5A), which indicated that the increase of lipid peroxidation was probably due to more ROS production. The above experimental results showed that high temperature caused $H_2O_2$ accumulation and peroxidation of perennial ryegrass plasma membrane, to a lesser degree in transgenic plants. Similarly, Guo et al. [13] found *Sm-MIR408* transgenic tobacco (*Nicotiana benthamiana*) decreased the accumulation of ROS under salt. Ma et al. [10] reported Arabidopsis plants overexpressing *miR408* accumulated less ROS when compared to WT under salt stress and oxidative stress, along with lower lipid peroxidation. These results further support a correlation between enhanced heat tolerance and elevated levels of miR408 in plants.

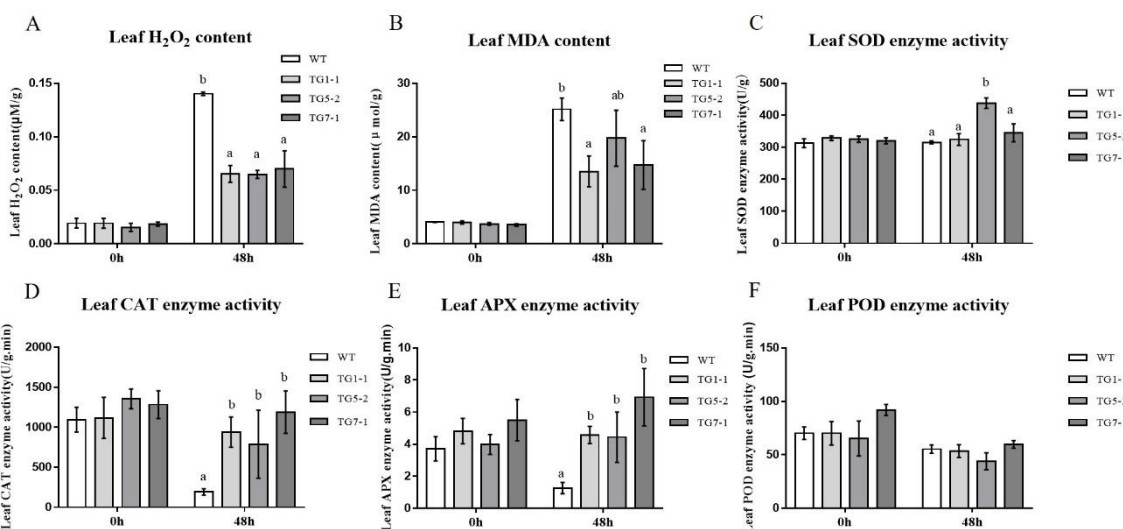

**Figure 5.** Hydrogen peroxide ($H_2O_2$) content, malondialdehyde (MDA) content, and antioxidant enzyme activities of transgenic lines (TG) in comparison with wild type (WT) control before and after heat stress treatment of 42 °C for 48 h in experiment 2. (**A**) Leaf $H_2O_2$ content, (**B**) leaf MDA content, (**C**) superoxide dismutase (SOD) activity, (**D**) catalase (CAT) activity, (**E**) ascorbate peroxidase (APX) activity, and (**F**) guaiacol peroxidase (POD) activity. 0 h, before heat stress; 48 h, 48 h after heat stress. Error bars represent standard error ($n$ = 5). Different lowercase letters indicate a significant difference between the wild type and each transgenic plant at $p < 0.05$.

Generally, ROS levels are well regulated by their degradation and generation rate as affected by the scavenging capacity of two systems, the non-enzymatic system (small molecular antioxidants) and enzymatic system (antioxidant enzymes) [36]. SOD constitutes the first line of the enzymatic defense system against ROS by dismutating $O_2^-$ to $H_2O_2$, and $H_2O_2$ is then finely regulated by CAT and an array of peroxidases, such as POD and APX [36]. There is plenty of evidence indicating that plant tolerance to various adverse environments is correlated to an increased capability to detoxify or scavenge ROS, and protection against oxidative stress is thought to be a critical component in determining the plant survival under heat stress [37]. We then examined the activities of SOD, CAT, APX, and POD of both TG and WT plants before and after heat stress in experiment 2. Under normal growth conditions, there was no significant difference in SOD enzyme activity between WT and TG plants, within the range of 312.87–328.56U/g FW. After 48 h of high temperature treatment, compared to the WT, the SOD activity of TG5-2 was higher, but not the other two TG lines (Figure 5C). Similarly, there was no significant difference in the CAT and APX enzyme activities between the leaves of WT and TG plants before heat stress (Figure 5D,E). After 48 h of high temperature treatment, the CAT and APX activity of WT plants was significantly inhibited and decreased to 193 U/g FW · min and 1.26 U/g FW · min, respectively, while their activities in the leaves of TG plants could still be relatively stable and maintained at a much higher level (Figure 5D, E). Interestingly, unlike CAT and APX, the POD enzyme activity was not different between WT and TG plants either before or after heat stress (Figure 5F). Overall, the transgenic perennial ryegrass could maintain higher antioxidative capacity under stress as manifested by higher activities of SOD, CAT, and APX. Similarly, tobacco plants overexpressing *miR408* had higher antioxidative capacity under salt stress with higher activities of SOD, POD, and CAT [13]. In addition, higher SOD activity under cold stress was also reported in plants overexpressing *miR408* [11]. Ma et al. [10] suggested that miR408 might increase the expression of some antioxidant genes, such as two copper/zinc SODs (*CSD1* and *CSD2*) via regulating copper homeostasis. Whether it is—the case in perennial ryegrass is not known yet, and further study would be needed.

## 4. Conclusions

Transgenic perennial ryegrass overexpressing *Os-miR408* showed improved heat stress tolerance when compared to WT, which could be partly due to the morphological changes of narrow leaves and smaller tiller angles, along with enhanced antioxidative capacity. This study could provide useful information for further understanding the molecular mechanism by which miR408 improves plant high-temperature tolerance, and offers new ideas and genetic resources for breeding heat-resistant cool-season turfgrass.

**Supplementary Materials:** The following are available online at https://www.mdpi.com/article/10.3390/agronomy11101930/s1, Table S1: Information of primers used in the study. Table S2: The sequence of the target gene. Table S3: The *PLASTOCYANIN* and *LAC3* mRNA cleave site of miRNA408.

**Author Contributions:** Conceptualization: K.W. and W.Z.; Methodology: K.W. and W.Z.; Supervision: K.W. and W.Z.; Data collection: G.T., N.H., T.S., Y.L., W.Y.; Formal analysis: G.T. and N.H.; Visualization: G.T. and N.H.; Writing—Original Draft: G.T. and N.H.; Writing—Review & Editing: G.T. and N.H. All authors have read and agreed to the published version of the manuscript.

**Funding:** This research was funded by National Natural Science Foundation of China, grant/award number: 31472140; Natural Science Foundation of Beijing Municipality, grant/award number: 6182025.

**Conflicts of Interest:** The authors declare no conflict of interest.

## Abbreviations

| | |
|---|---|
| SOD | superoxide dismutase |
| CAT | catalase |
| POD | guaiacol peroxidase |
| APX | ascorbate peroxidase |
| RWC | relative water content |
| EL | electrolyte leakage |
| ROS | reactive oxygen species |
| MDA | malondialdehyde |
| miRNA | microRNA |
| WT | wild type |
| TG | transgenic |

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
