# Peer review of "Ectopic Expression of Os-miR408 Improves Thermo-Tolerance of Perennial Ryegrass"

_agronomy, doi:10.3390/agronomy11101930_

Round 1

Reviewer 1 Report

 The Research indicate that the Ectopic Expression of Os-microrna408 Improves Thermo-Tolerance of Perennial Ryegrass,this work provide us the useful results as the climate change and global warming. The work can be published after the careful correction. 
1. Correct the temperature unit;
2. Table1 should move to supplimentary file;
3. Correct the unit in L186;
4. Correct H2O2 in L210, L216;
5. Figure 1 D and E, for the leaf width and tiller angle, only one line included?
6. The LAC3 mRNA cleave site of miRNA408  need to be present;
7.  L415 "Nicotiana  benthamiana" in italic;

Author Response

The Research indicate that the Ectopic Expression of Os-microrna408 Improves Thermo-Tolerance of Perennial Ryegrass, this work provide us the useful results as the climate change and global warming. The work can be published after the careful correction. 

Answer: Special thanks for the great suggestions. Those comments are very helpful for revising and improving our manuscript. According to your advice, we amended the relevant parts in the manuscript. Some of your questions were answered in details below.

  1. Correct the temperature unit;

Answer: Many of these superscript/subscript/italic format problems were most likely due to the reformatting of the editing office. We rechecked and corrected the formatting errors throughout the paper accordingly, such as the symbol of degree and the subscript of H2O2. The detailed changes were marked and highlighted in the manuscript. Thanks!

  1. Table1 should move to supplimentary file;

Answer: Good suggestion, and we moved Table 1 to supplementary file (Table S1).

  1. Correct the unit in L186;

Answer: Corrected as suggested.

  1. Correct H2O2 in L210, L216;

Answer: Corrected as suggested, and thanks!

  1. Figure 1 D and E, for the leaf width and tiller angle, only one line included?

Answer: Thanks a lot for the good question. The leaf width and tiller angle of transgenic plants (TG) for figure 1 D and E was an average of three lines, which represented the overall average level of over-expressing miR408 plants.

  1. The LAC3 mRNA cleave site of miRNA408  need to be present;
    Answer:

    We presented the LAC3 mRNA cleave site of miRNA408 in supplementary file (Table S3).

  2.  L415 "Nicotiana  benthamiana" in italic;

Answer: Corrected as suggested, and thanks!

Reviewer 2 Report

The authors presented a very nice and neat ms focusing on the function of miRNA408 in heat tolerance in perennial ryegrass. The experimental results are solid and supportive to their conclusion, and the ms is well written in general. I would recommend acceptance for publication after minor revision.

Here are my suggestions:

In the Result and Discussion section, we usu. list our result first and then discuss it. I do not feel comport when I read the section of 3.3.2 MiRNA408 increased plant antioxidant capacity. Please consider re-organized this section.

There are some grammar errors. Here are some examples:

  1. Latin name should be italicized. e.g. Line 54, for Lolium perenne, line 83 for Paeonia lactiflora, etc.
  2. Celsius symbols are wrong in the Introduction section but mostly correct in Materials and Methods. e.g. line 58-59.
  3. MicroRNA's targets are RNAs. Therefore, their targets should be italicized. E.g. Line 80-81.
  4. For H2O2, I saw that "2" is not in subscript. please correct them.

Author Response

Reviewer 2:

The authors presented a very nice and neat ms focusing on the function of miRNA408 in heat tolerance in perennial ryegrass. The experimental results are solid and supportive to their conclusion, and the ms is well written in general. I would recommend acceptance for publication after minor revision.

Here are my suggestions:

In the Result and Discussion section, we usu. list our result first and then discuss it. I do not feel comport when I read the section of 3.3.2 MiRNA408 increased plant antioxidant capacity. Please consider re-organized this section.

There are some grammar errors. Here are some examples:

Answer: Special thanks for the great suggestions. We revised the relevant parts in the manuscript accordingly. The detailed changes were marked and highlighted in the manuscript.

  1. Latin name should be italicized. e.g. Line 54, for Lolium perenne, line 83 for Paeonia lactiflora, etc.

Answer: Corrections were made accordingly, and thanks!

  1. Celsius symbols are wrong in the Introduction section but mostly correct in Materials and Methods. e.g. line 58-59.

Answer: Corrections were made accordingly.

  1. MicroRNA's targets are RNAs. Therefore, their targets should be italicized. E.g. Line 80-81.

Answer: Corrected as suggested, and please see the detailed changed in the revised manuscript.

  1. For H2O2, I saw that "2" is not in subscript. please correct them.

Answer: Corrections were made accordingly, and thank you!

5: Please consider re-organized this section (3.3.2 MiRNA408)

Answer: Thanks for your suggestion. We have re-organized the section of 3.3.2 “MiRNA408 increased plant antioxidant capacity”.

Reviewer 3 Report

Review of "Ectopic Expression of Os-microrna408 Improves Thermo-Tolerance of Perennial Ryegrass"

This paper describes improved thermo-tolerance of perennial ryegrass through the overexpression of Os-miR408. They show down-regulation of genes encoding plastocyanin and LAC3 in the transgenic plants compared to the WT grown under control conditions and compare phenotypes of WT and transgenic plants. This paper focuses on the transgenic lines and their morphological and physiological responses to show the improved heat tolerance.

It would be interesting to see how the expression of these genes (miR408, plastocyanin, LAC3) are regulated in response to heat stress in WT and transgenic plants, especially since you make a point of how they are differentially regulated in response to heat in different species. Maybe semi-quantitative RT-PCR after the heat treatments for both WT and transgenic lines?

How/why did you chose LAC3 as the target gene? Are there other LAC genes in perennial ryegrass that are affected by overexpressing Os-miR408? Are your primers specific for LAC3? Include the sequence of the gene you are monitoring and indicate if the primers are specific for LAC3.  

How often were the plants watered during heat stress? Heat stress can lead to drought stress, if plants are not watered adequately. Please indicate in the materials and methods.

For you semi-quantitative qRT-PCR analysis, I think it would be difficult to detect any differences for actin expression levels after 40 cycles?  

On page 11 line 46 you state, After 48 hours of high temperature treatment, compared to the WT, the SOD enzyme activity of TG plant leaves showed an overall higher trend  than the WT, especially TG5-2. Its SOD activity was 1.4 times of the WT (Figure 5C). NOTE: TG5-2 is really the only one that was different? The other two transgenic lines are very similar to the WT. Do you really think this represents an overall trend?

Some suggested corrections for this manuscript:

P4 Table

Plastocyanin Purpose for F and R primers is listed as RT-PCR reference?

Figure legend 2 – A WT and TG plants under normal conditions, (B,C) subjected to heat stress at 38 °C for 72 h, NEED SOMETHING MORE HERE, Is the second image a close up of 38 °C treatment, please indicate

Please check the degree symbol throughout the paper.

[] Please check for spaces before the [. Eliminate spaces after ] when followed by a period or comma.

Scientific names need to be italicized throughout the paper.

Generally, do not start sentences with ‘And’.

When using -1 to indicate per, should be a superscript (multiple places in the manuscript).

Be consistent – Use miR408 throughout the manuscript.

Check your gene/protein names and italicize when appropriate.

P2 line 65 vital (not vita)

P2 line 68 ‘biotic stresses (eliminate as well,)

L72 environmental stresses, such as (eliminate those of) iron deficiency

L80 also instead of or at the beginning of the line

L83 Instead, maybe: with much higher levels in …, suggesting the possible involvement of miR408 in heat stress responses and heat tolerance.

L87 In a genome wide profiling study of miRNAs in rice, miR408 expression levels were decreased in the heat tolerant cultivar N22, but increased in the susceptible cultivar Vandana.

P3 L91 species, experimental evidence is lacking on the function of miR408 in plant heat stress. The objective of this study was to examine… stress using previously established transgenic perennial ryegrass over-expressing the rice Os-miR408 gene [18]. Knowledge generated in this study will contribute valuable information towards understanding the role …

L120.. the plants were allowed to recover for seven days. To observe regrowth…

L126 growth conditions.

L131 After seven days of recovery, new leaves were counted..

L135 after removing the leaves from the incubator.

P4 Table

Plastocyanin Purpose for F and R primers is listed as RT-PCR reference

P5 L171 was measured as described by Blum ….

L179 The leaves were harvested at ….. treatment, and 0.2 g were ground in 10% ….

L187 Enzyme extracts were prepared using the method of Chaitanya et al. [25] with modifications.

L189 and then ground (eliminate ed)

L201 560 nm was measured …

L203 H202 use subscripts for 2 (here and throughout). Define ASA

P6 L212 Likewise, 100 mg of leaf tissues was frozen … and ground (no ed)

L231 RESULTS AND DISCUSSION (eliminate the T)

L239 angle, and several genes (instead of quite a few)

L241 More recently, miR167 has….

L243 MiR408 was suggested to be involved in anthocyanin …signaling [29]. Auxin responsive …

L247 further studies are warranted.

L248 architecture. Plants … habit often have a compact plant architecture… enhance photosynthetic efficiency….

L253 optimum temperature. Finer and smaller leaves are also thought … temperature [30]. Skip ‘as well”

L255 Thus we suggest that the morphological changes observed in miR408-over-expressing plants might enable the TG plants …

P7 L269 Genes encoding plastocyanin and LAC3 were down-regulated in miR408 over-expressing plants

L275 particularly in TG1-5? TG5-1?

L276 … proteins, which regulate a range of plant development processes (e.g., ..) and environmental stress responses, has been reported as a conserved regulatory pathway across different plant species [6,10].

L281 OsUCL30 were predicted targets of miR408 and have been validated by 5’ RACE [32]. Four plastocyanins .. were also validated as targets by investigating …

L284 are consistent with previous findings of the down-regulation of PLASTOCYANIN and/or LAC3 in miR408 over-expresssing lines [6,10].

P8 L290 Os-miR408 over-expressing perennial ryegrass were more tolerant to heat stress

L293 morphological (was misspelled)  Maybe rephrase?

L301 observed significant differences between …., with obvious wilting ..

L303 results were found

L319 [33]. The degree of damage mainly…

L330 Check to match figures numbers and what you say in the text. Figure 4B is chlorophyll, Figure A is Fv/Vm.  … Chlorophyll content of WT was significantly less than that found in TG5-2 and TG7-1.  

Author Response

Reviewer 3:

Special thanks for the great suggestions. According to your advice, we amended the relevant parts in the manuscript. Some of your questions were answered in details below.

Q1:It would be interesting to see how the expression of these genes (miR408, plastocyanin, LAC3) are regulated in response to heat stress in WT and transgenic plants, especially since you make a point of how they are differentially regulated in response to heat in different species. Maybe semi-quantitative RT-PCR after the heat treatments for both WT and transgenic lines?

Answer: Thanks a lot, and this is a very good advice. It would be better if we had revealed the response of these genes after heat stress. Here we mainly investigated the expression of the genes in order to verify the transgenic plants, and we will follow your great suggestion in our future work and pay more attention to how miR408 targets plastocyanin, LAC3c or other possible target genes and how they participate in regulating the heat stress response of plants in follow-up experiments.

Q2: How/why did you chose LAC3 as the target gene? Are there other LAC genes in perennial ryegrass that are affected by overexpressing Os-miR408? Are your primers specific for LAC3? Include the sequence of the gene you are monitoring and indicate if the primers are specific for LAC3.  

Answer: The LAC3 and plastosyanin were selected as target genes here based on both previous publications and the analysis of copper protein-related genes in the perennial ryegrass transcriptome data of our group (Wang et al., 2017) with the psRNA Target Tool (http:// Plantgrn.noble.org/psRNATarget/). The primers were designed according to the template sequences of LAC3 and plastosyanin from the transcriptome data. The sequence details were added in supplementary file (Table S2).

Q3: How often were the plants watered during heat stress? Heat stress can lead to drought stress, if plants are not watered adequately. Please indicate in the materials and methods.

Answer: Thanks for the question. During the heat stress, we placed a beaker with enough amount of water in the growth chamber to ensure that the temperature of the water is the same as the air temperature in the growth chamber. Based on our preliminary studies, the plants were watered every 12 hours to keep the soil moisture sufficiently and observed regularly to ensure that there was no obvious wilting. This description was added in the “materials and methods” part.

Q4: For you semi-quantitative qRT-PCR analysis, I think it would be difficult to detect any differences for actin expression levels after 40 cycles?  

Answer: Thank you for pointing this out. We double checked and the thermal cycler amplification condition of ACTIN gene was 35 cycles. We are sorry for the typo, and have made correction accordingly in the “materials and methods” part.

Q5: On page 11 line 46 you state, After 48 hours of high temperature treatment, compared to the WT, the SOD enzyme activity of TG plant leaves showed an overall higher trend than the WT, especially TG5-2. Its SOD activity was 1.4 times of the WT (Figure 5C). NOTE: TG5-2 is really the only one that was different? The other two transgenic lines are very similar to the WT. Do you really think this represents an overall trend?

Answer: Thanks a lot for the good question, and we very agreed that the statement about SOD activity was not precise.  We have re-written this part as following. “After 48 hours of high temperature treatment, compared to the WT, the SOD activity of TG5-2 was higher, but not the other two TG lines (Figure 5C).”

Q6: Some suggested corrections for this manuscript: (Eg: P4 Table Plastocyanin Purpose for F and R primers is listed as RT-PCR reference?)

Answer: Thanks for your suggestions. We have checked the manuscript again and corrected some errors according to the comments. The detailed changes were marked and highlighted in the manuscript.

Round 2

Reviewer 3 Report

Thank you for clarification and for corrections in the manuscript.